# Phage Digestion of a Bacterial Capsule Imparts Resistance to Two Antibiotic Agents

**DOI:** 10.3390/microorganisms9040794

**Published:** 2021-04-10

**Authors:** Cheng-Hung Luo, Ya-Han Hsu, Wen-Jui Wu, Kai-Chih Chang, Chen-Sheng Yeh

**Affiliations:** 1Department of Chemistry, National Cheng Kung University, Tainan 701, Taiwan; han19951101@gmail.com; 2Center of Applied Nanomedicine, National Cheng Kung University, Tainan 701, Taiwan; 3Department of Laboratory Medicine and Biotechnology, Tzu Chi University, Hualien 970, Taiwan; w200811@msn.com; 4Department of Laboratory Medicine, Buddhist Tzu Chi General Hospital, Hualien 970, Taiwan

**Keywords:** *Acinetobacter baumannii*, bacteriophage therapy, antibiotic-resistance, phage-antibiotic synergy, lytic phage

## Abstract

Bacteriophages are viruses that infect bacteria, replicating and multiplying using host resources. For specific infections, bacteriophages have developed extraordinary proteins for recognizing and degrading their host. Inspired by the remarkable development of viral proteins, we used the tail fiber protein to treat multiple drug-resistant *Acinetobacter baumannii*. The tail fiber protein exhibits polysaccharide depolymerases activity which specifically degrades exopolysaccharide (EPS) during the phage–host interaction. However, EPS-degraded cells are observed altering host susceptibility to bacterial lysis peptide, the endolysin-derived peptide. Notably, endolysin is necessary in the process of progeny liberation by breaking the bacterial cell wall. Surprisingly, peeling the EPS animated host to resist colistin, the last-resort antibiotic used in multidrug-resistant Gram-negative bacteria infection. Tail fiber-modified cell wall reduces colistin attachment, causing temporary antibiotic-resistance and possibly raising clinical risks in treating multiple drug-resistant *A. baumannii*.

## 1. Introduction

Prior to the discovery and development of antibiotics, bacteriophages were an important therapeutic agent for bacterial infection disease, but suffer from critical drawbacks of narrow spectrum and high specificity [1]. However, increasing antibiotic abuse led to selective stress, driving the development of multiple drug resistant (MDR) ‘superbugs’, raising the urgent need to screen new antibiotics or develop alternative therapeutic strategies [2]. Over the past two decades, the US Food and Drug Administration (FDA) and European Medicines Agency have only approved two new antibiotic classes to treat Gram-positive bacterial infections [3]. Parallel to the emergence of various strategies to develop new antibiotics [4,5,6], new attention has turned to clinical phage therapy applications, with the establishment of the first phage therapy center in North America, the Center for Innovative Phage Applications and Therapeutics (IPATH).

Bacteriophages are ubiquitous, and are present in volumes at an order of magnitude greater than bacterial counts [7]. The development of ‘superbugs’ has driven renewed interest in therapeutic phage applications for its specific lysis and self-replication properties. Specific applications have included palliative use of MDR *Acinetobacter baumannii*, safe and effective comprehensive treatment in MDR ESKAPE, and even the development of personalized phage products [8,9,10]. Moreover, clinical trials of low concentration (10^2^ plaque forming unit) phage therapy in the European Clinical Trials database showed no significantly adverse impact when compared against standard antibiotic treatments for burn patients [11]. However, some instances of failed phage treatment were found to exhibit phage-resistant bacterial isolates. Similar to the antibiotic-resistant mechanisms, phage-resistant behavior has also been observed in some basic research.

Mechanisms for phage resistance include restriction modification (R-M) systems and bacterial immunity via the CRISPR-Cas system [12]. Based on the phage infection processes, the bacterial host develops various resistance mechanisms for blocking phage absorption, invasion, and propagation [13]. Different from the R-M and CRISPR-Cas system, strategies for rejecting initial recognition are relatively complex and feature various resistance mechanisms through bacterial hosts [14]. Better known mechanisms include modification of extracellular components, membrane proteins, or bacterial flagella as the first entrance of phage initial attachment [15]. Nevertheless, clinical trials have found that ‘phage-antibiotic synergy’ (PAS) provides better outcomes than either individual antibiotic or phage therapy [16]. In addition, phage-derived proteins have also been used for clinical treatment of infection. For example, the polysaccharide depolymerase was used to attenuate bacterial cell walls or destroy biofilms. Clinical applications have been developed based on how lytic phages use their holins and endolysins to escape from the host [17]. More recent efforts have addressed bacterial selection to treat drug-stress or phage infection [18]. Selectivity against the expression of virulence factors to resist phage infection may attenuate host defense to antibiotics, promoting the development of PAS therapy and realizing the mechanism of action and stoichiometry [19].

Previously, we identified 8 lytic podophages for clinical MDR *A. baumannii* [20]. The podophages were characterized as having fast host recognition, rapid multiplication, and host lysis capability, attracting interest for advanced studies in rapid recognition and lysis nature. Endolysin from the phage ϕAB2 was cloned and shown to dominate the lysis process for breaking bacterial cells. Subsequently, analysis of the endolysin was shown to have a high degree of antibacterial activity against both gram-positive and gram-negative bacteria, which was found that *C*-terminal amphipathic peptides containing basic amino acids interact with negatively charged membrane [21]. The tail fiber (TF) with function of capsular depolymerase is another attractive virion protein. Some applications of phage-derived depolymerase have been used in penetration of bacterial cell wall, reducing impairment of immunological response, or against bacterial biofilm [22,23]. Furthermore, depolymerase therapy as an antibiotic was shown to be in robust health of mice study [24]. In our case, two TF proteins were previously identified and found to dominate the host range determination [25]. A structural study showed TF possesses a bacterial receptor binding domain and an exopolysaccharide (EPS) depolymerization domain [26]. Based on these findings, we propose a combination strategy of co-treating MDR *A. baumannii* with antiobiotics (or phage endolysin derived peptide, P3) and TF molecules. However, no additive or synergistic effect on clinically isolated MDR *A. baumannii* was observed. We surprisingly observed a temporary increase in resistance to colistin, used as the last-resort antibiotic for MDR Gram negative bacterial infection. Evaluation of the phage attacking behavior suggests the temporary increased resistance may be associated with an increased risk of drug-resistance during PSA in clinical MDR *A. baumannii* treatment.

## 2. Materials and Methods

### 2.1. Materials

All reagents were of analytical purity. Iron(III) acetylacetonate (Fe(C_5_H_7_O_2_)_3_, 99.9%), trioctylamine (TOA, [CH_3_(CH_2_)_7_]_3_ N, 98%), and oleic acid (OA, CH_3_(CH_2_)_7_CHCH-(CH_2_)_7_COOH, 90%) were used as purchased from Sigma-Aldrich (St. Louis, MO, USA). Chloroform (CHCl_3_, 99.8%) was obtained from MERCK (St. Louis, MO, USA). Ethanol (C_2_H_5_OH, 99.9%) was purchased from J.T. Baker (Loughborough, Leicestershire, UK).

### 2.2. Electron Microscopy 

Morphology and characterization of the nanoparticles and bacteria were monitored by transmission electron microscopy (TEM, Hitachi H-7500) Surface analysis of the bacteria was observed by high-resolution scanning electron microscope (HR-SEM, JEOL JSM-7001F).

### 2.3. Preparation of 22 nm Fe_3_O_4_ NPs 

For preparation of 22 nm Fe_3_O_4_ nanoparticles, 1.4 g iron(III) acetylacetonate, 0.52 mL oleic acid, and 20 mL trioctylamine were mixed in a round-bottomed flask, and heated at 150 °C for 30 min to dewater, after which the temperature was lowered to 120 °C. A suction pump was used to degas for 30 min, after which the temperature was increased to 305 °C (rise rate: 1.5 °C/per min) under full argon conditions. The reaction was maintained at 305 °C for 30 min and then cooled to room temperature. Iron oxide nanoparticles (IONPs) were collected by centrifugation (8000× *g* rpm, 10 min) and then washed using toluene at least 3 times, and then dried for further characterization.

### 2.4. Preparation of Tail Fiber Decorated 22 nm Fe_3_O_4_ NPs 

The as-prepared oleic acid capped IONPs (1 mg) were dispersed in chloroform at a concentration of 10,000 ppm. DSPE-PEG-COOH (purchased from Biochempeg Scientific Inc., Watertown, MA, USA) was dissolved in pure water. An equal weight ratio of IONP and DSPE-PEG-COOH was mixed and sonicated for 10 min at room temperature. The water dispersed IONPs were washed two times with water and redispersed in water for further use. The purified tail fiber (TF) protein (described in detail later) was anchored on DSPE-PEG decorated IONP using an amine functional group, assisted by EDC (1-Ethyl-3-[3-dimethylaminopropyl]carbodiimide hydrochloride) and Sulfo NHS (N-Hydroxysulfosuccinimide) to form an amide bond. TF anchored IONPs were washed two times with water and redispersed in a phosphate buffer for further use.

### 2.5. Bacterial Strains and Growth Conditions

Two bacterial strains *A.b*-M2835 (Phage2 Host) and *A.b*-54149 (Phage6 Host) were previously isolated from clinical samples. Bacteria were cultivated in Luria–Bertani (LB) broth or LB agar (Difco Laboratories, Detroit, MI, USA) at 37 °C.

### 2.6. Tail Fiber Protein Expression and Purification

The DNA fragments encoding the tail fiber proteins were cloned into pET30a expression vectors (Novagen, Merck KGaA, Darmstadt, Germany). After the sequences were confirmed, the vectors were transformed into *E. coli* BL21(DE3) (Novagen, Merck KGaA, Darmstadt, Germany). The cells were cultured in Luria-Bertani (LB) medium supplemented with 50 μg/mL kanamycin at 37 °C. Once the cell density reached OD600 of 0.4–0.6, bacterial cells were induced with 0.1 mM isopropyl-_L_-_D_-thiogalacto pyranoside at 37 °C for 4 h, and the cells were harvested by centrifugation (6000× *g* rpm) at 4 °C for 30 min. The cell pellet was suspended in 10 mL of lysis-equilibration-wash (LEW) buffer containing 50 mM NaH_2_PO_4_/300 mM NaCl (pH 8.0), disrupted by sonication and centrifuged at 10,000× *g* for 15 min to remove debris. Crude supernatant was loaded onto Protino Ni-TED packed columns (MACHEREY-NAGEL, Düren, Germany) equilibrated with LEW buffer. The fractions were eluted with the elution buffer containing 50 mM NaH_2_PO_4_/300 mM NaCl/250 mM imidazole (pH 8.0). Active fractions were pooled and dialyzed against the elution buffer and concentrated by an Amicon Ultra-0.5 centrifugal filter (MILLPORE, Bedford, MA, USA). The concentration of each purified protein was determined by Bradford assay using bovine serum albumin as a standard.

### 2.7. Preparation of FITC Modified Colistin

Colistin and fluorescein isothiocyanate (FITC) were purchased from Sigma-Aldrich. The isothiocyanate group of FITC spontaneously reacts with amines to form a stable thiourea bonding. In brief, colistin was dissolved into a borate buffer (sodium borate 100 mM, pH 9.0). FITC was immediately dissolved into DMSO (1 mg/mL) before use. The colistin and FITC solutions were immediately mixed at a molar ratio of 10:1 and protected from light. The reaction was allowed to stand for 2 h at room temperature or overnight at 4 °C in the dark. Unreacted molecules were filtered by a dialysis membrane and the reaction buffer was replaced with water.

### 2.8. Measurement of Fluorescence Intensity

Colistin attachment efficiency was assessed by using ImageJ software. Fluorescent images were taken using laser confocal microscopy (C2 confocal microscope, Nikon, Tokyo, Japan). TF treated (or nontreated) bacteria adhered by colistin-FITC conjugate was measured under the same parameters. Average fluorescence intensity was statistically analyzed using pixel analysis for the regions of interest.

### 2.9. Statistical Analysis

Statistical analysis was evaluated using paired Student’s *t*-test and analysis of variance (ANOVA).

## 3. Results and Discussion

### 3.1. Depolymerase-Treated Bacteria Persisting Antibiotic Resistance

We used two distinct TF sources, i.e., phage ϕAB2 and phage ϕAB6, for the treatment of individual bacterial hosts. A preliminary test for bacteria treatment evaluated morphological changes following TF treatment of the corresponding host. The surface of TF-treated bacteria changed from rough to smooth (Figure 1a) in both host bacteria treated by phage ϕAB2-tail fiber (TF2) and phage ϕAB6-tail fiber (TF6). Lee et al. provided a structural description of the enzyme activity from TF6, which hydrolyzes the extracellular exopolysaccharide (EPS) into oligosaccharide units [26]. The hydrolyzed products showed that EPS comprises - 3)-β-GalNAcp-(1 → 3)-[β-Glcp-(1 → 6)]-β-Galp-(1 -units, and the major repeat contains pseudaminic acid. Since EPS has been characterized as being associated with MDR and resistance to adverse environments [27], we then submitted the TF-treated bacterial cells to antibiotic challenge.

Given the surface smooth after TF treatment, the bacteria were expected to become drug sensitive. We initially postulated the TF surface treatment may alter some functions of the biofilm-related efflux pump, and tetracycline-TF synergy was tested based on the drug-resistance mechanism [28,29]. However, tetracycline resistance was found to be indistinguishable from that of the TF-treated bacterial cells (Figure 1b). Other antibiotics were also tested based on potential diffusion efficacy after removing the EPS barrier, but results were disappointedly ineffectual (Appendix A).

### 3.2. Peeling the EPS Has the Surprising Effect of Protecting the Bacterium against Colistin

Despite the ineffectualness of TF peeling in producing antibiotic susceptibility, we propose that the exposed cell membrane may more susceptible to endolysin derived peptide (P3 peptide) [21] than is the EPS protected membrane. However, results surprisingly indicate EPS peeled bacterial cells exhibit greater resistance to P3 peptide than the nontreatment one (Figure 2a). This unexpected result led us to associate P3 peptide with colistin, the last-resort antibiotic for Gram negative bacterial infection. The two molecules show structural similarity of positively charged amphipathic peptide between P3 peptide and colistin. Similar to the PAS test, we combined colistin and P3 peptide to evaluate the combined use of antibiotic and phage peptide. As expected, a significantly synergistic bactericidal effect was observed under colistin and P3 peptide co-treatment (Figure 2b). However, co-incubating *A. baumannii* with colistin and the TF brought on a result similar to the protective effect (Figure 2c). Mechanisms for colistin-resistance have been previously studied in various bacteria, including *Klebsiella pneumoniae*, *Pseudomonas aeruginosa*, and *A. baumannii* [30]. The majority of such mechanisms showed that bacteria use various strategies to protect the cell membrane from colistin adherence by positively modifying the lipopolysaccharide or overexpressing the membrane associated proteins. One distinct EPS negative mutation revealed that structurally dependent adherence is necessary for initially efficient attachment during colistin attack [31]. The increased drug-resistance following TF treatment may also result from the same mechanism.

### 3.3. Increased Colistin Resistance Is Not Due to Tail Fiber–Colistin Interaction

To rule out the possibility of TF–colistin molecular interaction leading to reduced colistin attachment, we anchored TF on the surface of 22-nm iron oxide nanoparticles (IONP) to mimic phage particles without infectious capability (the relevant identifications are shown in Appendix A). As shown in Figure 3a, bacterial cells were treated by IONP-TF. After treatment, bacterial cells were collected by centrifugation, followed by magnetic removal of IONP-TF. Initially, we confirmed the TF enzyme activity by mixing IONP-TF with the host bacteria and observed the cellar morphology under electron microscopy (Appendix A). After identifying the inoculated bacterial number, 10^5^ cells were cultured with colisitn at various concentrations using TF6 and its host. Results show a significant protective effect on bacteria after TF removal of the host EPS, based on colony forming unit (CFU) counting (Figure 3b). Furthermore, we conjugated fluorescein isothiocyanate to colistin to observe the antibiotic attaching efficiency. Fluorescent images show the reduced colistin attack on cell membranes (Figure 3c). On the other hand, we also observed that IONP-TF particles did not stably adhere to bacterial cells (Appendix A), which may explain partial survival of the bacteria during phage invasion.

### 3.4. Colistin Resistance and Persistence of Both Lytic Phage and Host Bacteria

Lytic phages infect bacteria by injecting viral genome, hijacking the host system, replicating virion components, and breaking out of the bacterial cells. Once the bacterium is hijacked by the phage, survival is impossible under systemic regulation from the phage genome. However, the current phage may randomly preserve part of the host by peeling the EPS using non-infective (i.e., genome ejected) phage particles. Similar phage–host interaction was previously reported in K1 phage that specifically infects *E. coli* K1 strain [32]. Investigation on the bacterial mutant of capsular reduction and phage defect of depolymerase activity, capsular binding is essential, but degradation is not required or even harmful during infection process. However, degradation may help the phage penetration of the dense capsular polysaccharide of the wild-type bacteria. To evaluate the TF in our phage–host interaction, we conducted a one-step growth test to assess the phage infection efficiency (Figure 4), finding that EPS peeled cells reduce phage infection efficiency and produce fewer phage particles, which corresponds to other EPS degradable phages in the initial infection step [13]. Observation from the IONP-TF–bacteria interaction, particles showed quickly degrading and leaving the EPS. Previous studies also showed nonspecific capture using IONP-TF as the affinity binding property for the detection of *A. baumannii* [33]. Taken together, phage particles behave quickly binding, degrading, infection, and desorption during invasion process. The remaining capsid keep degrading the non-infected bacteria based on the depolymerase function of TF. Furthermore, EPS peeled cells show strong resistance to endolysin-derived peptide, which may accidentally break bacterial cells if endolysin leaks from the phage infected host. The interaction accidently sustains both the lytic phage and host population, and helps part of the host to temporarily resist phage infection and endolysin destruction (Figure 5).

## 4. Conclusions

The incident results in bacterial protection by TF from endolysin destruction and induces colistin resistance, which behaves similarly to endolysin functional peptide. This contradicts previous findings that phage resistance may attenuate bacterial drug resistance, and offers a reasonable explanation for virulent phage resistance without altering the host genome for phage and bacterial persistence. The results may also suggest consideration of avoiding combination phage and colistin in clinical use.

## Figures and Tables

**Figure 1 microorganisms-09-00794-f001:**
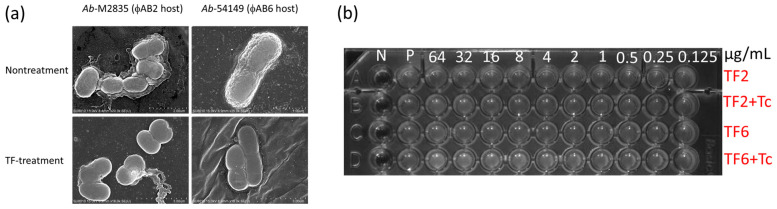
Scanning Electron Microscope (SEM) pictures and antibiotic susceptible test results of tail fiber (TF) treated bacteria. (**a**) The bacterial hosts for phageAB2 and phageAB6 were treated using the corresponding TF proteins. The effective treatment showed a significant morphological change under SEM observation. (**b**) Test of tetracycline susceptibility shows that TF treatment does not alter drug-resistance. Various concentrations of TF were marked on the 96-well dish, with 10 μg/mL tetracycline used as culture medium.

**Figure 2 microorganisms-09-00794-f002:**
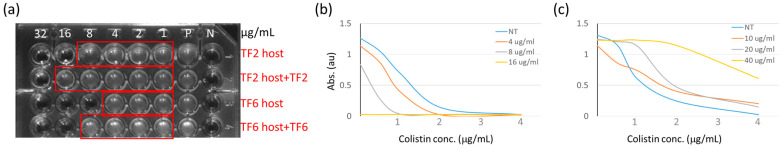
Reducing bacterial susceptibility to endolysin-derived P3 peptide and colistin by tail fiber (TF) treatment. (**a**) TF treated bacteria show resistance to P3 containing culture medium. Various concentrations of P3 peptide were marked on a 96-well dish. TF treated hosts were found to be more resistant to P3 peptide than the untreated control. (**b**) Turbidity test (595 nm absorbance) to determine bacterial growth showed an additive effect in the suppression of *A. baumannii*. Curves of various colors represent the different P3 dosages which are showing on the indicated concentrations. (**c**) Protective effects were observed when presenting TF protein in *A. baumannii* culture medium. The dosages for TF protein are presented according to the mark for different color.

**Figure 3 microorganisms-09-00794-f003:**
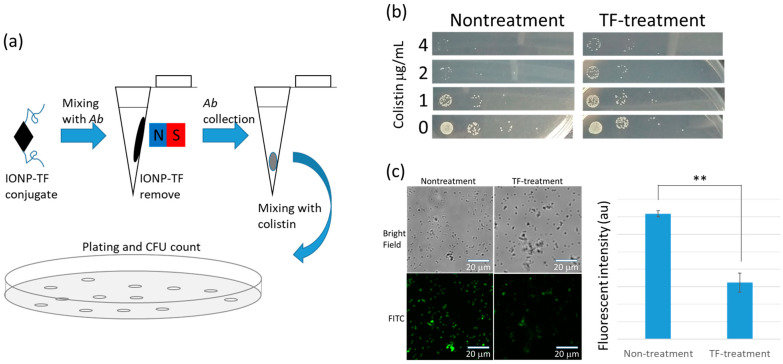
Mimicry of tail fiber (TF) protein effectiveness in phage initial infection and test of TF-induced colistin resistance. (**a**) TF conjugated iron oxide nanoparticles (IONP) were used to treat bacteria and magnetically removed before colistin challenge. (**b**) Determination of colistin resistance by counting the colony forming units. The resistance capability of the TF-treated bacteria is nearly an order of magnitude greater than control group. (**c**) Fluorescent images of colistin–fluorescein isothiocyanate (FITC) conjugate show that nontreatment bacteria exhibited higher colistin affinity than the TF-treated host, and the fluorescent intensity was significantly higher than the TF-treated group.

**Figure 4 microorganisms-09-00794-f004:**
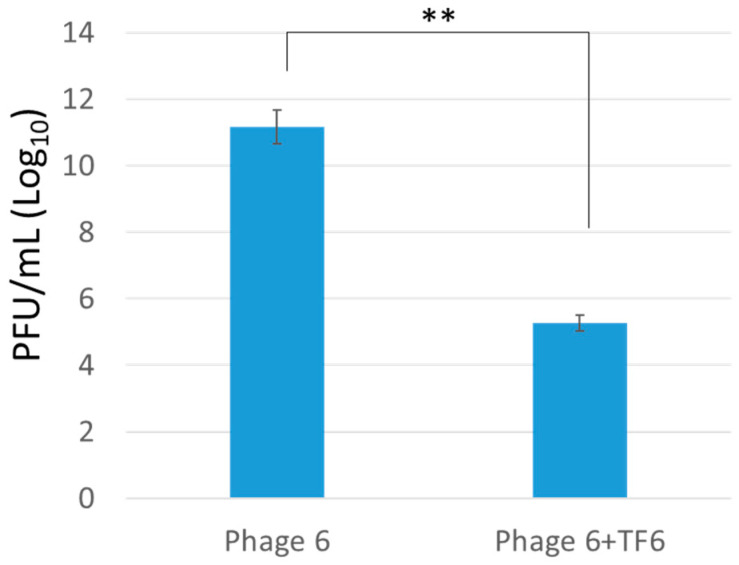
Phage propagation test on exopolysaccharide (EPS)-degraded bacteria. Here, the figure shows the comparison for phage propagation efficiency test in control bacterial host and tail fiber-treated bacteria. The tail fiber treated bacteria significantly reduced phage propagation efficiency.

**Figure 5 microorganisms-09-00794-f005:**
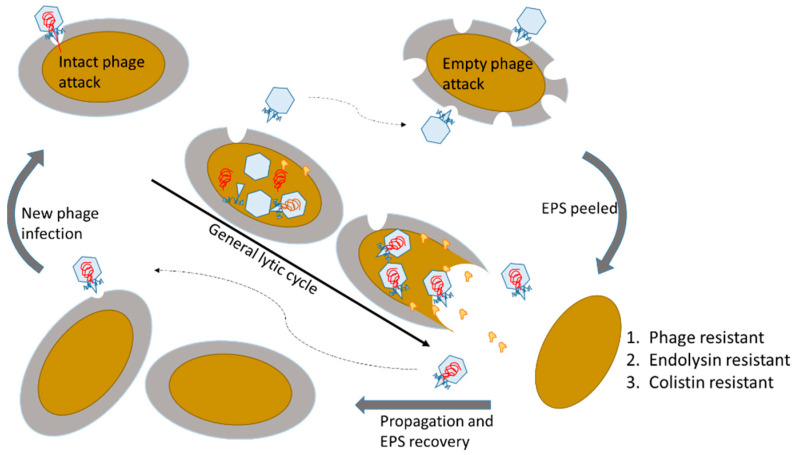
Scheme for the proposed resistance of exopolysaccharide (EPS) degraded bacteria. The initial step for EPS degradable phages is critically adhering the host EPS and degrading the host EPS to adsorb the host membrane, followed by injecting viral genome into the host cells. Loss of EPS reduces the phage binding efficiency. On the other hand, we also discovered that the EPS peeled bacterial cells strongly resist phage endolysin and colistin by reducing attachment of the positively charged amphipathic peptide.

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
