# Peer review of "Phage Digestion of a Bacterial Capsule Imparts Resistance to Two Antibiotic Agents"

_microorganisms, 2021, doi:10.3390/microorganisms9040794_

Round 1
Reviewer 1 Report
This is a fairly straightforward study that will be of interest to the phage therapy field. The basic result is that a phage-encoded depolymerase can be used to strip a bacterial capsule (at least I think it is a formal capsule) from a bacterium (Acinetobacter b.) One would think that stripping the capsule would render the bacterium increasingly susceptible to agents that act on or require access to the outer, but they find the opposite. This puzzle is studied in some detail to understand whether changes in attachment are involved.
To the extent I understand things, the work is sound and the results interesting.
I have some comments about matters that should be easily corrected.
1) The ms uses language about evolution and co-evolution. There is no evolution in the paper, and any reference to evolution should be omitted, lest it cast doubt on the entire paper. Everything here is a phenotypic effect -- there is no propagation of strains to view how they evolved in response to treatment. The title is nonsense, for example. A better title would be simply: 'Phage digestion of a bacterial capsule imparts resistance to two antibiotic agents.'
2) The ms does a poor job of referencing the literature. There is a lot of work on phage depolymerases as antibiotics (in vivo); a review by Sutherland and Azeredo is a good starting point. There is also a precedent for loss of a capsule inhibiting binding of a phage that uses the capsule for adsorption: Pelkonen, S., Aalto, J., Finne, J., 1992. Differential activities of bacteriophage depolymerase on bacterial polysaccharide: binding is essential but degradation is inhibitory in phage infection of K1-defective Escherichia coli. J. Bacteriol. 174, 7757–7761. (This paper provides a precedent for part of Fig. 5, for example.)
3) Line 163: the specificity of the tail fibers (TF) for host is only demonstrated one way. If it was not demonstrated the other way, then the statement should be qualified. But it is known that other tailspike depolymerases are specific to capsule type, not necessarily to other aspects of the bacterium, so the present example fits with what is known.
4) Line 179. 'expected to reverse drug insensitivity thereby attenuating the bacterial surface protection' is a convoluted way of saying 'expected to become drug sensitive.'
5) Line 182: 'identical to' is more correctly 'indistinguishable from'
6) Line 186: change the subheading to emphasize the puzzle: 'peeling the EPS has the surprising effect of protecting the bacterium against colistin' or something similar.
7) Fig. 2 legend should provide more detail about the figures. Since there are two agents being combined, the key within the figures should indicate that the doses are of P3.
8) Line 243: 'However, the current phage may randomly preserve ...' is a statement of teleology that imparts a purpose to the phage's behavior yet has little foundation in evolutionary biology. It contributes to the general impression that evolutionary theory is misunderstood in the paper.
9) Line 248: 'This suggests that IONP-TF quickly degrades and leaves EPS' is confusing. Reword.
10) LIne 254: 'coevolution results in' is another example of the misuse of evolutionary biology here.
Jim Bull
Author Response
Point 1: The ms uses language about evolution and co-evolution. There is no evolution in the paper, and any reference to evolution should be omitted, lest it cast doubt on the entire paper. Everything here is a phenotypic effect -- there is no propagation of strains to view how they evolved in response to treatment. The title is nonsense, for example. A better title would be simply: 'Phage digestion of a bacterial capsule imparts resistance to two antibiotic agents.'
Response 1: We thank the suggestions and comments from reviewer. We have modified the title as ‘Phage digestion of a bacterial capsule imparts resistance to two antibiotic agents’ and the modifications have been highlighted in title page and supplementary materials part. Additionally, we also omitted all wording of ‘evolution’ in original descriptions.
Point 2: The ms does a poor job of referencing the literature. There is a lot of work on phage depolymerases as antibiotics (in vivo); a review by Sutherland and Azeredo is a good starting point. There is also a precedent for loss of a capsule inhibiting binding of a phage that uses the capsule for adsorption: Pelkonen, S., Aalto, J., Finne, J., 1992. Differential activities of bacteriophage depolymerase on bacterial polysaccharide: binding is essential but degradation is inhibitory in phage infection of K1-defective Escherichia coli. J. Bacteriol. 174, 7757–7761. (This paper provides a precedent for part of Fig. 5, for example.)
Response 2: We thank the suggestions and comments from reviewer. We have added some introductions related to depolymerase therapy and applications of holing or endolysin. The starting statement is a review by Sutherland and Azeredo (please see line 59, page 2). Similar phage proteins with depolymerase function and its applications were introduced in line 73, page2. The suggested literature from J. Bacteriol. (1992) 174, 7757–7761. has been cited in the paragraph 3.4., please see line 244, page 6.
Point 3: Line 163: the specificity of the tail fibers (TF) for host is only demonstrated one way. If it was not demonstrated the other way, then the statement should be qualified. But it is known that other tailspike depolymerases are specific to capsule type, not necessarily to other aspects of the bacterium, so the present example fits with what is known.
Response 3: We thank the suggestions from reviewer. we have omitted the statement for the specificity which have been known as in other phage.
Point 4: Line 179. 'expected to reverse drug insensitivity thereby attenuating the bacterial surface protection' is a convoluted way of saying 'expected to become drug sensitive.'
Response 4: We thank the suggestions from reviewer. We have modified the sentence as reviewer’s suggestion. Please see line 178, page 4.
Point 5: Line 182: 'identical to' is more correctly 'indistinguishable from'.
Response 5: We thank the suggestions from reviewer. We have modified the sentence as reviewer’s suggestion. Please see line 182, page 4.
Point 6: Line 186: change the subheading to emphasize the puzzle: 'peeling the EPS has the surprising effect of protecting the bacterium against colistin' or something similar.
Response 6: We thank the suggestions from reviewer. We have modified the sentence as reviewer’s suggestion. Please see line 185, page 5.
Point 7: Fig. 2 legend should provide more detail about the figures. Since there are two agents being combined, the key within the figures should indicate that the doses are of P3.
Response 7: We thank the suggestions and comments from reviewer. We have indicated the dosage of P3 in Fig. 2 (b) and tail fiber protein in Fig. 2 (c). The illustration sentence was highlighted in line 211-214, page 5.
Point 8: Line 243: 'However, the current phage may randomly preserve ...' is a statement of teleology that imparts a purpose to the phage's behavior yet has little foundation in evolutionary biology. It contributes to the general impression that evolutionary theory is misunderstood in the paper.
Response 8: We thank the suggestions and comments from reviewer. The original description has been omitted. According to the suggestions from point 2, literature from J. Bacteriol. (1992) 174, 7757–7761. is a good referenced for explaining binding and degradation behaviors during phage-host interaction. Binding is essential for host recognition, and degradation may be not critical but help to penetrate the dense capsular polysaccharide. In our case, the degradation surprisingly robust the host in colistin challenge. The description has been highlighted in line 247-251, page 6.
Point 9: Line 248: 'This suggests that IONP-TF quickly degrades and leaves EPS' is confusing. Reword.
Response 9: We thank the comment from reviewer. The sentence has been modified as ‘Observation from the IONP-TF-bacteria interaction, particles showed quickly degrading and leaving the EPS.’ in line 254, page 6.
Point 10: Line 254: 'coevolution results in' is another example of the misuse of evolutionary biology here.
Response 10: We thank the comment from reviewer. The wording 'coevolution results in' has been omitted, and give the paragraph to explain the phage-host interaction as ‘Taken together, phage particles behave quickly binding, degrading, infection, and de-sorption during invasion process. The remaining capsid keep degrading the non-infected bacteria based on the depolymerase function of TF.’ Please see line 257-259, page 6.
Reviewer 2 Report
1) Please, the authors should comply with the International rules of Bacterial taxonomy; this meaning that in the present manuscript the bacterial names should be written in italics.
2) Interesting results concerning Acinetobacter baumanii and phage therapy; the present results support the idea that such therapy should include a cocktail of phages, or else phage-derived lysins, and not colistin whatsoever, but any other antibiotic. To my opinion, the paper merits publication provided the number 1 is taken care of.
Author Response
Point 1: Please, the authors should comply with the International rules of Bacterial taxonomy; this meaning that in the present manuscript the bacterial names should be written initalics.
Response 1: We thank the comment from reviewer. We have corrected the mistake in the manuscript.
Point 2: Interesting results concerning Acinetobacter baumanii and phage therapy; the presentresults support the idea that such therapy should include a cocktail of phages, or elsephage-derived lysins, and not colistin whatsoever, but any other antibiotic. To myopinion, the paper merits publication provided the number 1 is taken care of.
Response 2: We thank the comments from reviewer.